# Machine learning driven biomarker selection for medical diagnosis

**Divyagna Bavikadi**[1]*, **Ayushi Agarwal**[1], **Shashank Ganta**[1], **Yunro Chung**[2,3],
**Lusheng Song**[2], **Ji Qiu**[2], **Paulo Shakarian**[1]

1 Fulton Schools of Engineering, Arizona State University, Tempe, Arizona, United States of America,
2 Biodesign Center for Personalized Diagnostics, Arizona State University, Tempe, Arizona, United States
of America, 3 College of Health Solutions, Arizona State University, Phoenix, Arizona, United States of
America

* dbavikad@asu.edu

HONG KONG

**Peer Review History:** PLOS recognizes the
benefits of transparency in the peer review
process; therefore, we enable the publication of
all of the content of peer review and author
responses alongside final, published articles.
The editorial history of this article is available
here: https://doi.org/10.1371/journal.pone.
0322620

**Data availability statement:** Relevant data are
within the paper and its Supporting Information
files. The data underlying the results presented
in the study are also available from
https://pubmed.ncbi.nlm.nih.gov/33108201/

## Abstract

Recent advances in experimental methods have enabled researchers to collect data on
thousands of analytes simultaneously. This has led to correlational studies that asso-
ciated molecular measurements with diseases such as Alzheimer's, Liver, and Gas-
tric Cancer. However, the use of thousands of biomarkers selected from the analytes is
not practical for real-world medical diagnosis and is likely undesirable due to potentially
formed spurious correlations. In this study, we evaluate 4 different methods for biomarker
selection and 5 different machine learning (ML) classifiers for identifying correlations—
evaluating 20 approaches in all. We found that contemporary methods outperform pre-
viously reported logistic regression in cases where 3 and 10 biomarkers are permit-
ted. When specificity is fixed at 0.9, ML approaches produced a sensitivity of 0.240 (3
biomarkers) and 0.520 (10 biomarkers), while standard logistic regression provided a
sensitivity of 0.000 (3 biomarkers) and 0.040 (10 biomarkers). We also noted that causal-
based methods for biomarker selection proved to be the most performant when fewer
biomarkers were permitted, while univariate feature selection was the most performant
when a greater number of biomarkers were permitted.

## Introduction

Recent advances in experimental methods have enabled researchers to collect data on thou-
sands of analytes (biological analytes) simultaneously [1,2]. This has led to correlational stud-
ies that associated these molecular measurements with diseases such as Alzheimer's [3], Liver
[4], and Gastric Cancer [5]. However, it is generally considered undesirable to use thousands
of biomarkers selected from the analytes for medical diagnosis for several reasons. First, large
numbers of biomarkers increase the likelihood of spurious correlation. Second, the use of
many biomarkers increases model complexity and hinders the interpretability of results. Fur-
ther, from a practical standpoint, the use of fewer biomarkers is preferable from the stand-
point of creating cost-effective diagnostic products.

**Funding:** The author(s) received no specific funding for this work.

**Competing interests:** No authors have competing interests.

As a result, previous studies have conducted two operations in tandem: the selection of candidate biomarkers thought to be associated with a given disease individually and the identification of correlations between the combination of selected candidate biomarkers and the target medical condition. The most commonly reported methodology in the literature has been logistic regression, often accompanied by a variant of univariate feature selection [6–8]. This paper looks to augment existing work by studying the effect of the feature selection method and model type. In particular, we examine causal-based feature selection [9] and a variety of machine-learning approaches, including gradient-boosted decision trees and neural networks. In all, we study 20 different combinations of feature selection and classification models in tests where the number of biomarkers $K$ is restricted to a set of values 1,3,4,10,15,30 on a gastric cancer dataset that includes measurements from 3440 biological analytes [10]. We perform a cross-validation study and report results on training and test sets as well as examine hyperparameter sensitivity for the causal-based approaches. We found that contemporary machine learning methods outperform previously reported logistic regression in these experiments. When specificity is fixed at 0.9, ML approaches produced a sensitivity of 0.240 (3 biomarkers) and 0.520 (10 biomarkers), while standard logistic regression provided a sensitivity of 0.000 (3 biomarkers) and 0.040 (10 biomarkers).

The rest of the paper is organized as follows: We first provide a brief overview of related work, a description of the gastric cancer dataset, and machine learning methods. This is followed by reporting of the experimental results on the gastric cancer dataset and associated discussion. Finally, we conclude by discussing our findings.

## Gastric cancer dataset

The dataset [10] used for the biomarker discovery contains information on 100 samples, each of which is associated with a case or control indicating the presence or absence of gastric cancer. The dataset is balanced with 50 samples labeled case and 50 samples labeled control. The age and gender of the samples are matched between cases and controls. Each instance is represented by 3440 corresponding molecular measurement values, which are used to assess the risk of gastric cancer and provide insight into the disease. The measurement values range from 0.00 to 260.65 with a median of 1.00. Molecular measurements were noted with IgG and IgA antibodies against the same set of proteins. The dataset contains data on clinical features, antibody reactions against *Helicobacter pylori* proteins, and demographic variables. Using the Nucleic Acid Programmable Protein Array (NAPPA) technology, the study assessed humoral responses to 1527 proteins or almost the whole *H. pylori* proteome. The total set of proteins nearly composes a complete *H. pylori* proteome. Measurement values were assessed on seropositivity. Seropositivity was defined as the median normalized intensity $2 \leq$ on NAPPA. Table 1 shows the breakdown of the dataset.

**Table 1. Breakdown of gastric cancer dataset.**

| Data | Samples | Analytes data | Quantity |
|---|---|---|---|
| Total Samples | 100 | Total measurements* | 3440 |
| Cancer Cases | 50 | Organism: *H. Pylori* | 3054 |
| Cancer Controls | 50 | Organism: EBV | 178 |
| | | Organism: Streptococcus_gallolyticus | 92 |
| | | Organism: Fusobacterium_nucleatum | 84 |
| | | Organism: Other (≤5 occurrences) | 32 |

* indicates that it includes *IgG* and *IgA* antibodies.

Up to our knowledge, this dataset [10] has not been explored for different biomarker selection methods at the time of writing this paper.

For the training data, each sample has a vector of real values associated with each analyte measurement and a ground truth that indicates the actual presence of the disease to distinguish between gastric cancer patients from healthy controls.

## Machine learning and feature selection methods

### Overview of approaches

We employ a two-step process for each method: feature selection and classification, and will discuss each in turn. We will use the symbol $K$ (let $K \in \{1, 3, 4, 10, 15, 30\}$) to denote the maximum number of biomarkers permitted after the feature selection step. The best $K$ biomarkers are used to then classify a sample. We also explore the effect of binarizing biomarker inputs—the intuition being that rather than considering the biomarker measurement directly, we only consider if the biomarker exceeds some threshold $\gamma$ ($\gamma \in \{0.6, 1.0, 1.4, 1.8\}$), which is specified as a hyperparameter. Note that even though the measurements range up to 260.65, most values are around 1.00. Due to this distribution of values, we accordingly set values for $\gamma$.

The whole process can be viewed in Fig 1. Considering the size of the data, we use Leave-one-out cross-validation (LOOCV). We compute the causal metric (replaced by a univariate selection method when needed) to select the top $K$ biomarkers that have the highest causality measures (for univariate selection, the chi-squared test is based on higher scores of the relationship between the feature and the target variable- which correspondingly have a low p-value). We then use $K$ biomarker measures to train and test the ML models. We then report the results for various hyperparameter settings. We also parallelize the causal computation for every analyte $i$ in Equation 1 for runtime efficiency.

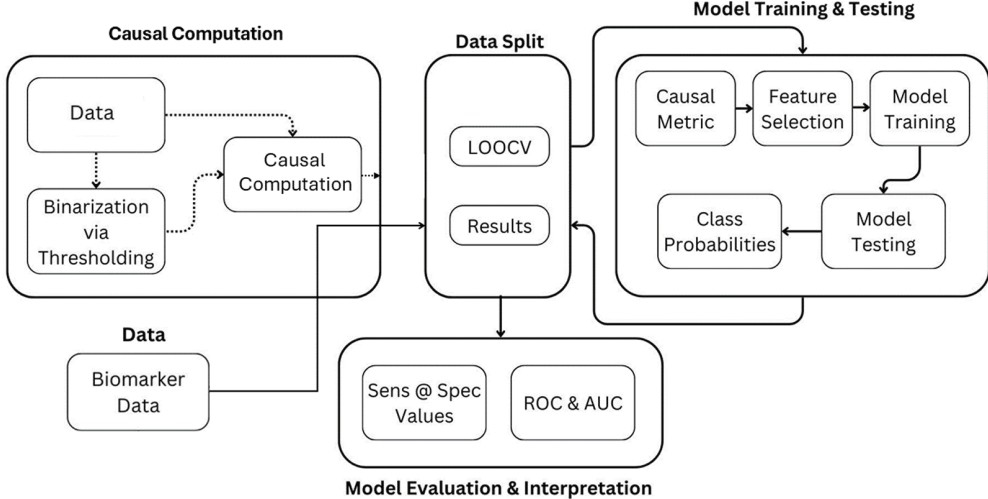

**Fig 1. Overall workflow.**

## Feature selection methods

We consider two types of feature selection methods: the univariate selection and the causal metric. Univariate feature selection evaluates the strength of the relationship between the feature and the response variable. In this paper, we use chi-square statistic-based univariate feature selection method. By contrast, the causal-based method examines the effect of a single analyte based on other analytes that may have a co-occurring measurement. A contribution of this work is an adaption of the causal measure of Kleinberg et al. [11] for biomarker selection. While Kleinberg et al. [11] computes causality as the average increase in the probability of the effect when the cause is present, here we propose a new metric based on the intuition of Gardner et al. [12] but adapted for biomarker selection as follows:

$$causal(i) = \frac{\sum_{j \in R_i} f(i,j) - f(\neg i, j)}{size(R_i)} \tag{1}$$

Here we still examine the average increase of a function when the biomarker is present based on co-occurring biomarkers. However, unlike in Kleinberg et al. [11] we do not use probability, but a measure more tuned to our domain. In Equation 1 the symbol $causal(i)$ is the causal metric for the analyte $i$, $R_i$ indicates the set of analytes that are related to the analyte $i$, and $f$ indicates the measure calculated based on the product of sensitivity and specificity (also known as $s2$ metric) for every pair of a analyte $i$ and its related analyte $j$. This makes it more suitable for the kind of protein biomarkers used from the dataset. We provide details as to how we derived this measure in the Supporting information.

Here in Table 2 is an example for calculation of the causal metric on a sample dataset of 4 biomarkers for 4 instances. The data is binarized with the threshold of $\gamma = 1.4$ and these binarized values are shown in the brackets in Table 2. The $s2$ metric is computed for all biomarkers and the ones with a value greater than the average $s2$ metric are considered as seen in the 6th row in Table 2. The related biomarkers for each biomarker is computed when there is at least one overlap of a case sample where the biomarker value is greater than the threshold $\gamma$. Finally, the causal metric is computed using Equation 1. Here, $B3$ and then $B1$ would have been picked during feature selection process. Note that in this example, few biomarkers got a $NaN$ value like for biomarker $B2$ because fo lack of related biomarkers but similar case is not commonly observed when implementing on the complete gastric cancer dataset. Further details about the causal metric can be found in the supporting information.

**Table 2. Causal metric computation on sample dataset.**

| label | B1 | B2 | B3 | B4 |
|---|---|---|---|---|
| 0 | 1.28 (0) | 3.24 (1) | 0.32 (0) | 4.56 (1) |
| 1 | 1.73 (1) | 0.21 (0) | 2.12 (1) | 1.12 (0) |
| 1 | 0.50 (0) | 1.62 (1) | 0.67 (0) | 0.45 (0) |
| 0 | 2.31 (1) | 0.82 (0) | 1.32 (0) | 3.65 (1) |
| s2 | 0.25 | 0.25 | 0.5 | 0 |
| > avg(s2) | 1 | 1 | 1 | 0 |
| $R_i$ | B3 | 0 | B1 | 0 |
| causal(i) | 0.25 | NaN | -0.25 | NaN |

## Machine learning classification methods

We examine four machine learning methods: logistic regression (LR), random forest (RF), deep multi-layer perceptron (MLP), gradient-boosted decision trees (GBT) [13], and XGBoost (XGB) [14]. The intuition for using logistic regression is to establish it as a baseline as it was used in previous biomarker studies [7,15], random forest for its ability to provide accurate results with minimal hyper-parameter tuning, a Deep Neural Network (DNN) due to their state-of-the-art performance in a variety of other tasks, and two variants of boosted trees which have been shown to provide state-of-the-art performance on tasks involving tabular data. For the DNN, we employ a dense, multi-layer perceptron with 4 layers, Rectified Linear Unit (ReLU) as an activation function, and a softmax output layer using the PyTorch [16] software package. For the boosted decision trees, we use the Scikit-learn implementation of gradient-boosted trees and the standard implementation of XGBoost. Summaries of these methods, along with hyperparameter settings can be found in Supporting information.

## Results

### Setup

We conducted experiments using an NVIDIA GTX1080 (2560 cuda cores, 10 Gbps memory speed). For evaluation, we used leave-one-out cross-validation (LOOCV) and examined values for Area Under the Curve (AUC) for both training and test data, as well as sensitivity on the test data with specificity fixed at 0.8 and 0.9 (sensitivity at specificity of 0.8 (*Sen*@80) and sensitivity at specificity of 0.9 (*Sen*@90)). These metrics are selected based on standards employed in assessing diagnostic biomarkers; it also helps us have an overall understanding of performance across multiple confidence thresholds as well as judge the degree to which the model can discriminate between case and control. Evaluation of experiments is conducted on this standard based on other factors such as models, and hyperparameters. Throughout the discussion, we will treat logistic regression with univariate selection as the baseline, as logistic regression was employed in prior work [7,8].

We use the xgboost python package for the XGBoost model and the sklearn package for the other models in our experiments. The sklearn package also has the leave-one-out cross-validation split inherent to it. The model hyperparameters mentioned in Table 3 are considered as the default setting unless specified.

### Selection of 3 biomarkers

Overall, the most performance in terms of test AUC was observed for the deep neural multilayer perceptron (MLP) classifier with causal metric for biomarker selection, which outperformed the baseline by 0.114, shown in Table 4. For Sensitivity at specificity of 0.9, XGB with causal metric (as seen in Fig 2) outperformed the baseline (as seen in Fig 3) by 0.240.

**Table 3. Model hyperparameters used for each model.**

| MLP | | XGB, GBT | |
|---|---|---|---|
| hidden_layer_sizes: | 256,128,64,32 | max_depth: | 2.0 |
| activation: | relu | learning_rate: | 1.0 |
| random_state: | 1.0 | n_estimators: | 10.0 |
| | | random_state: | 0.0 |
| **LR** | | **RF** | |
| solver: | lbfgs | n_estimators: | 10.0 |

**Table 4. Results for 3 biomarkers using 5 models with causal-based and univariate feature selection.**

| Model | Method | Train AUC | Test AUC | Sen@90 | Sen@80 |
|---|---|---|---|---|---|
| **MLP** | Univariate | 0.937 | 0.581 | 0.080 | 0.140 |
| | Univariate(B) | 0.738 | 0.527 | 0.000 | 0.000 |
| | Causal | 0.720 | **0.695** | 0.220 | 0.420 |
| | Causal(B) | 0.774 | 0.588 | 0.200 | 0.300 |
| **XGB** | Univariate | 0.969 | 0.613 | 0.200 | 0.260 |
| | Univariate(B) | 0.754 | 0.538 | 0.000 | 0.000 |
| | <u>**Causal***</u> | <u>**0.719***</u> | <u>**0.633***</u> | <u>**0.240***</u> | <u>**0.480***</u> |
| | Causal(B) | 0.611 | 0.463 | 0.200 | 0.340 |
| **LR** | Univariate | 0.699 | 0.612 | 0.000 | 0.180 |
| | Univariate(B) | 0.756 | 0.560 | 0.000 | 0.000 |
| | Causal | 0.678 | 0.510 | 0.180 | 0.280 |
| | Causal(B) | 0.771 | 0.594 | 0.200 | 0.200 |
| **GBT** | Univariate | 0.984 | 0.571 | 0.120 | 0.280 |
| | Univariate(B) | 0.738 | 0.527 | 0.000 | 0.000 |
| | Causal | 0.722 | 0.659 | 0.140 | 0.420 |
| | Causal(B) | 0.613 | 0.496 | 0.220 | 0.360 |
| **RF** | Univariate | **0.997** | 0.558 | 0.120 | 0.200 |
| | Univariate(B) | 0.736 | 0.620 | 0.060 | 0.080 |
| | Causal | 0.719 | 0.593 | 0.120 | 0.120 |
| | Causal(B) | 0.662 | 0.583 | 0.180 | **0.540** |

(B) dictates using binarized data; **Bolded** values dictate better performance; <u>**Underlined***</u> values dictate best performance

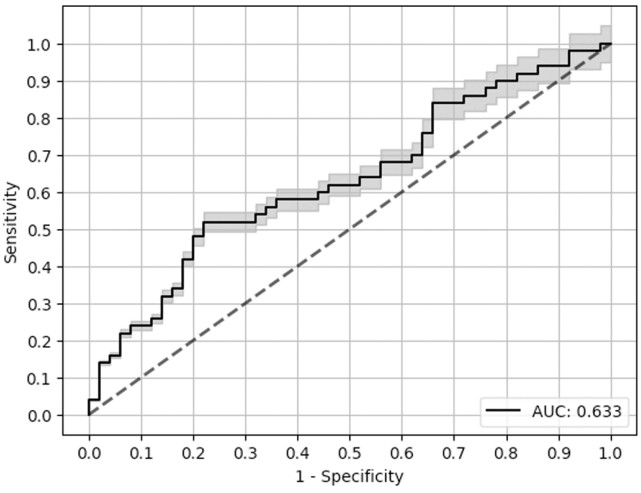

**Fig 2. ROC curve for XGB model with causality measure (3 biomarkers).**

The error bars in the Receiver Operating Characteristic (ROC) curves are for the confidence bands of 5% error. Notably, the use of causality feature selection improved performance irrespective of classifier, providing a minimum improvement of 0.120 (binarized) over univariate feature selection for each classifier for *Sen*@90 (Table 4). Comparable results were noted for Sensitivity when Specificity was set to 0.8 along with test AUC.

We note that training AUC was strongest for random forest with univariate selection with a value of 0.997—however, this drops to 0.558 for testing. This is surprising, as random forest generally does not overfit [17]; however, it may indicate that univariate feature selection

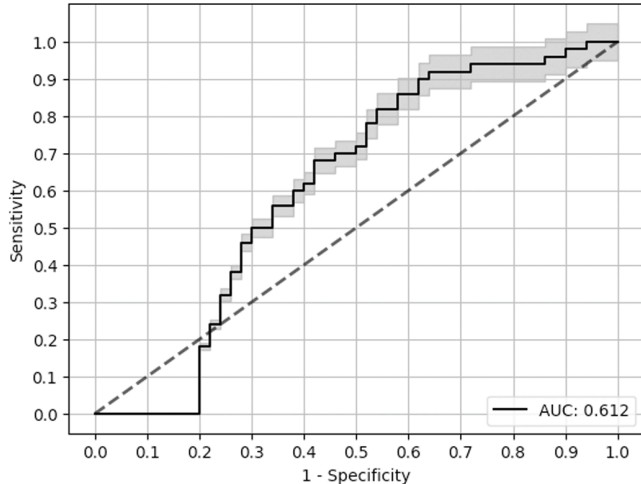

**Fig 3. ROC curve for the baseline (3 biomarkers).**

may cause overfitting when used in more complex models—as we observed the large discrepancies between training and testing AUCs when univariate feature selection was used in all cases except logistic regression. On the other hand, the average drop for the causality measure is 0.118 and a maximum of 0.186 while there is an average drop of 0.260 and a maximum of 0.439 for univariate feature selection which indicates a possibility of overfitting caused when causality is ablated.

## Selection of 10 biomarkers

On the other hand, the best-performing model, with respect to test AUC, was MLP with univariate feature selection, which outperformed MLP with causality measure by 0.286, shown in Table 5. Furthermore, GBT with univariate feature selection (as seen in Fig 4) reported the highest sensitivity at a specificity of 0.9, that is 0.520 while GBT with causality measure reported sensitivity at a specificity of 0.9 as 0.22. Also, the baseline (as seen in Fig 5) gave a moderate test AUC of 0.599 but a low $Sen@90$ value. The error bars in the ROC curves are for the confidence bands of 5% error. We found that, with a high number of biomarkers, univariate feature selection seems to be performing well with respect to test AUC compared to the causality measure for all methods by a minimum of 0.025 (binarized) and 0.029 (non-binarized).

For a higher number of biomarkers, a more generic method like univariate seems to suffice. While increasing the historical data might help improve the performance of other approaches, the less data-hungry causal approach already performs well without inconsistent sensitivity at a specificity of 0.9, 0.8.

## Hyperparameter study

As shown in Tables 4 and 5, a few methods were classified based on the binarization of biomarker values before model training indicated by B; for example, Causal(B) means causality method with binarized inputs. We discretize all input measurements for a given sample based on a threshold $\gamma \in \{0.6, 1.0, 1.4, 1.8\}$. Tables 4 and 5 are for optimal hyperparamter settings among the chosen values. However, it is important to note that there is little variance in

**Table 5. Results for 10 biomarkers using 5 models with causal-based and univariate feature selection.**

| Model | Method | Train AUC | Test AUC | *Sen@90* | *Sen@80* |
|-------|--------|-----------|----------|----------|----------|
| **MLP** | Univariate | 1.000 | 0.669 | 0.140 | 0.340 |
| | Univariate(B) | **0.926** | **0.764** | **0.480** | **0.480** |
| | Causal | 0.909 | 0.551 | 0.200 | 0.260 |
| | Causal(B) | 0.918 | 0.478 | 0.120 | 0.200 |
| **XGB** | Univariate | 0.998 | 0.701 | 0.300 | 0.420 |
| | Univariate(B) | 0.890 | 0.684 | 0.460 | 0.460 |
| | Causal | 0.816 | 0.575 | 0.200 | 0.340 |
| | Causal(B) | 0.879 | 0.659 | 0.220 | 0.360 |
| **LR** | Univariate | 0.811 | 0.599 | 0.040 | 0.180 |
| | Univariate(B) | 0.878 | 0.746 | 0.460 | 0.480 |
| | Causal | 0.734 | 0.569 | 0.080 | 0.220 |
| | Causal(B) | 0.830 | 0.681 | 0.320 | 0.360 |
| **GBT** | **Univariate*** | **1.000*** | **0.721*** | **0.520*** | **0.620*** |
| | Univariate(B) | 0.919 | 0.746 | 0.480 | 0.500 |
| | Causal | 0.852 | 0.588 | 0.220 | 0.260 |
| | Causal(B) | 0.875 | 0.540 | 0.180 | 0.220 |
| **RF** | Univariate | **0.999** | 0.649 | 0.140 | 0.380 |
| | Univariate(B) | 0.926 | 0.708 | 0.420 | 0.440 |
| | Causal | 0.894 | 0.594 | 0.140 | 0.260 |
| | Causal(B) | 0.904 | 0.538 | 0.120 | 0.320 |

(B) dictates using binarized data; **Bolded** values dictate better performance; <u>**Underlined***</u> values dictate best performance

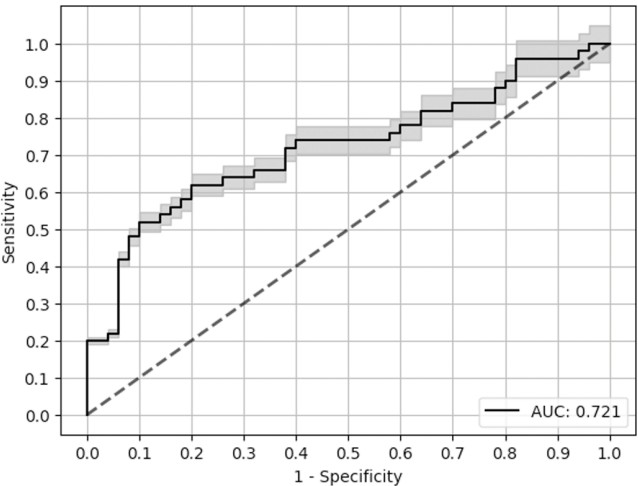

**Fig 4. ROC Curve for GBT model with univariate feature selection (10 Biomarkers).**

AUCs for most thresholds except for 1.0, showing the stability of the selected biomarkers as seen in Fig 6 for XGB with causal metric for 3 biomarkers. The figure shows the ROC curves across different threshold values (including 0.6,1.0,1.4,1.8). Similar trend is observed for other methods as well, where the threshold value of 1.0 typically results in lower test AUCs and the most optimal threshold is 1.4. Also, consistency is observed in the frequency of biomarker selection. Furthermore, by raising the value of $K$ significantly, we get diminishing returns, suggesting a saturation point to pick the number of biomarkers, $K$.

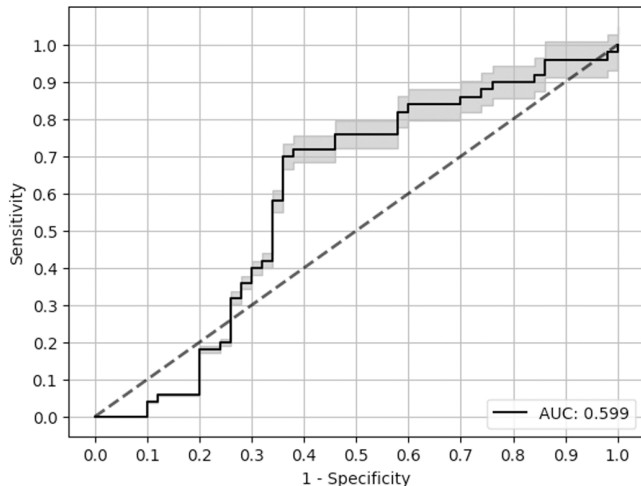

**Fig 5. ROC curve for the baseline (10 biomarkers).**

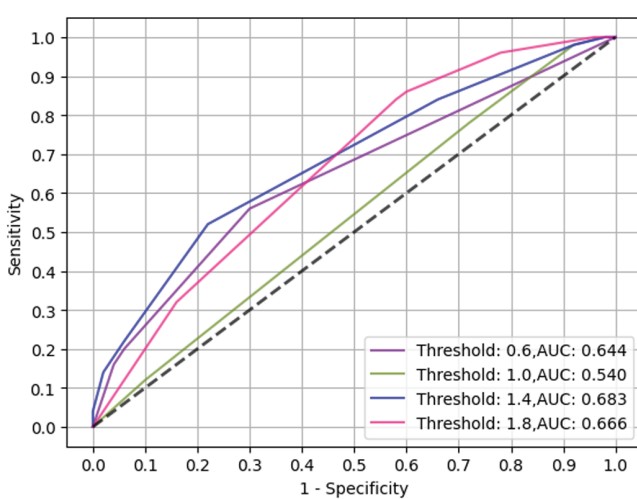

**Fig 6. Hyperparameter Sensitivity.** ROC Curve with multiple thresholds($\gamma$) for XGB model with causal-based biomarker selection.

We found the biomarkers: Epstein-Barr virus capsid protein *BFRF*3, Holliday junction resolvase-like protein *HP*0334, hypothetical proteins *HP*1029, *HP*0386, *HP*0273, *HP*1065, hydrogenase expression/formation protein *HP*0898, acyl-CoA thioesterase *HP*0496 IgA antibodies and Epstein-Barr nuclear C-terminal Glutathione S-Transferase *EBNA cGST* IgG antibody, to be few of the most frequently selected biomarkers related to gastric cancer for the threshold 1.4. Fig 7 shows the high frequency of biomarkers for various threshold values. As seen in the figure, for low values of *K*, most biomarkers appear in above 90% of all folds when evaluating with LOOCV, therefore supporting the stability of the model. These are the biomarkers that were consistently picked by the causality measure.

Notably, the test AUC increases with *K* and saturates after *K* = 10 as seen in Fig 8. However, *K* had a limited impact on the biologically relevant *Sen*@90 measure. Initially, increasing the value of *K* increased the test AUC by the magnitude of 0.2. As we gradually increased *K*, the

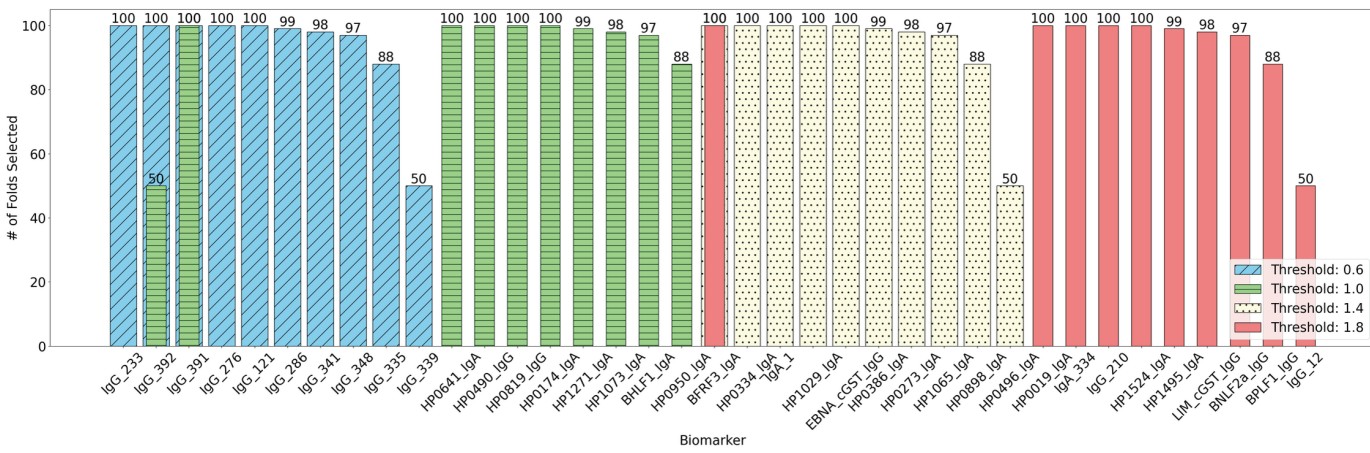

**Fig 7. Hyperparameter sensitivity.** Frequency of Selected Biomarkers, where *K* = 10 for multiple thresholds($\gamma$).

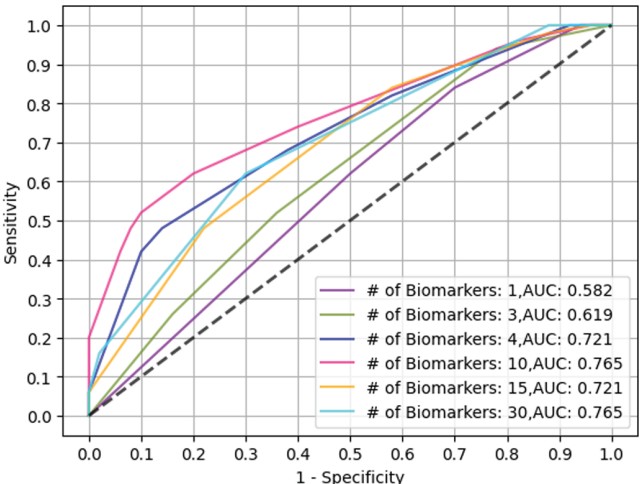

**Fig 8. Hyperparameter sensitivity.** Effect of *K* for threshold 1.4 for GBT model with univariate selection.

test AUC levels out to a certain range, around 0.7 but the *Sen*@90 measure tends to get more sparse. We see diminishing returns by adding any more number of biomarkers. This relation has relevance based on the target application desired to make inexpensive diagnostic kits.

## Discussion and limitations

We use biomarker measures for cancer prediction and leverage the causality measure to select causal biomarkers. We see the effects of ablating away causality measure with univariate feature selection in Table 4. We observe a higher AUC and consistent sensitivity values for the causal method as we decrease the number of biomarkers, and these benefits go away otherwise. This will be beneficial when applying this method to the industry considering the computational power and being less expensive as the method performs better with less number of biomarkers. This approach can also be applicable to similar domains for other disease prediction. Additionally, the experiments with the causal metric can be extended by adding a combinatorial way of picking the ranked causal biomarkers.

For the secondary analysis, we recorded the increase in the probability of cases with the presence of the causal-based biomarkers as a function of $K$ as seen in Fig 9 for the threshold of 1.4. The error plots are of a confidence band of 5% error. Similar to the performance of test AUC as a function of $K$, the probability of increase escalates till $K = 10$, but then drops and saturates with a further increase in $K$. This also supports our finding of benefits with a causal-based method for lower biomarkers. Specifically, for a lower number of biomarkers, the causal-based method gave a probability of increase up to 84.375%, while on the other hand, for a higher number of biomarkers than $K = 10$, there is a decrease in the probability of cases with the presence of those biomarkers up to 11.11%. Note that, here, the biomarkers are the most frequent with higher causal scores across all folds in LOOCV. Future inquiry of interest also includes empirical experimentation to validate ML models as well as, specific biological testing that can be performed on particularly selected causal biomarkers.

Our method for selecting a few biomarkers had the best performance, however, we found a limitation in that it didn't perform as well as other baselines when allowed for over 3 times more biomarkers. We saw evidence of overfitting for most approaches with univariate in 3 biomarker settings with a drastic drop from train to test AUC, in particular, MLP for 10 biomarkers as well. Given the dataset size, it is known that approaches like MLP will overfit. However, we dint observe this for the causal-based method on a low number of biomarkers.

Experiments with the binarized inputs were conducted to get better causal explanations as it allows you to localize the sources of causality, by having clear contributions of which biomarkers are selected versus not. For a threshold, we can precisely extract biomarkers that are causal compared to all potential biomarkers. However, on average, it does almost the same as non-binarized experiments with respect to test AUC. Moreover, note that Fig 9 illustrates that we indeed obtained biomarkers that are causal.

## Related work

Machine learning models, such as logistic regression, have been utilized with biological data for association purposes. In Islam et al. [8], the correlation coefficients of three biomarkers: body temperature, heart rate, and probable blood glucose level, were evaluated and associated

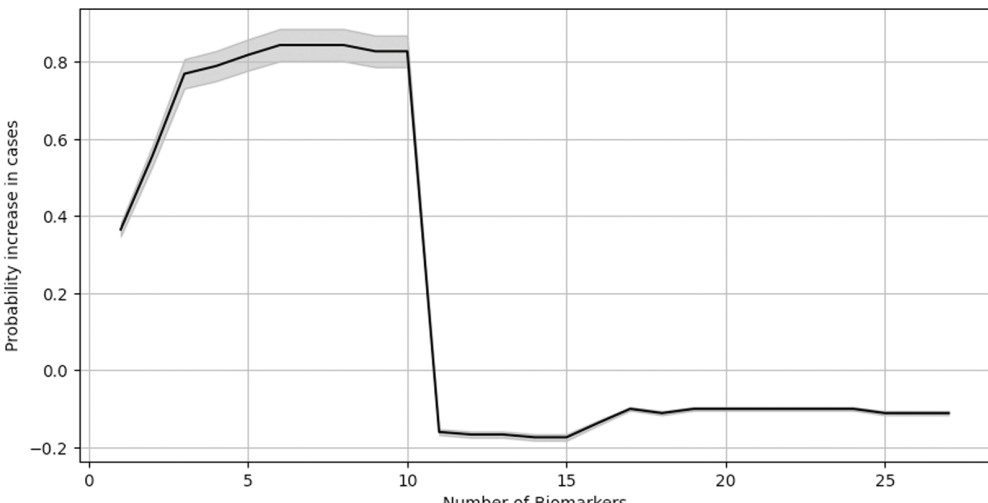

**Fig 9. Increase in probability of cases (cancer) with the presence of causal-based biomarkers as a function of $K$.**

with malaria detection using logistic regression. Similarly, in Direkvand-Moghadam et al. [7], univariate logistic regression demonstrated a substantial association between female sexual dysfunction and biomarkers, such as age, gravidity, and menarche age. Additionally, in Bursac et al. [6], the application of feature selection prior to model training showed the potential to maintain confounding variables, especially when dealing with macro biological data sets. Note that none of this prior work conducts an analysis of various machine learning classifiers, such as gradient-boosted trees or neural networks with causal-based and feature selection methods.

More specifically, machine learning models paired with feature selection for disease detection have proved significantly beneficial. Various hybrid optimization methods were used in combination with ML models like Decision Tree, Logistic Regression, Random Forest,etc. [18–20] and evaluated on cancer data based on sensitivity, specificity, and ROCs akin to our evaluation. However, note that they were applied to datasets with features of a maximum of 30, while we observe a dataset with 100 times more analyte measures. Typically, for a disentangled feature selection for pre-classification, univariate feature selection and dimensionality reduction methods are used for high dimensional datasets [21–23]. Also, a gradient-boosting decision tree, logistic regression was used with multivariate analysis and other feature importance methods on gastric cancer data based on a limited number of characteristics [24,25], we use a univariate analysis to be computationally less intensive over a huge number of analyses. Note that Recursive Feature Elimination methods are computationally expensive with growing data size while model-entangled feature importance (GBM Importance, Lasso) will not be a fair comparison to the causal-based method. Considering our dataset size and to give a fair comparison while retaining original feature measures, we set univariate selection as a baseline. In Sorino et al. [26], numerous machine learning techniques similar to ours, such as random forest classifier and boosted tree classifier, with cross-validation were used to diagnose non-alcoholic fatty liver disease. Similarly, in Díaz Álvarez et al. [27], a feature selection, evaluated on chi-squared statistic was paired with a Naive Bayes classifier to aid the diagnosis and classification of neuro-degenerative disorders. Moreover, vision-based machine learning techniques such as convolutional neural networks have been applied to a wide variety of medical diagnostic use cases [28–32]), even applied for gastric cancer image data [33]. Some techniques used k-fold cross-validation with chi-square test combined with other hybrid nature-inspired feature selection methods and found XGBoost to be effective when used on images of chest CT scans [34], although the authors concentrate on leveraging radiological image features and converge to minimum 90 features. Such diagnosis based on imagery would be complementary to biomarker-based diagnosis. However, to our knowledge, the application of such techniques to the use of biomarkers, specifically a large number of proteins, for the purposes of medical diagnosis has not been studied in the literature. We also allow for a parameter $K$ to limit the number of biomarkers and find that our approach performs well for as low as 3 biomarkers.

The concept of causal-based methods, such as the one apparent in our findings, has been used in a variety of medical applications [11]. For example, in Richens et al. [35], the application of causal machine learning effectively increased clinical accuracy from the top 48% to the top 25% of doctors. However, to date, such methods have not been combined with recent advances in biomarker experimentation [36] for medical diagnosis based on biomarker measurements. We focus on the narrow area of leveraging biomarkers for cancer prediction

## Conclusion

In this paper, we use a causality measure to select biomarkers paired with ML-based classifiers on a gastric cancer dataset for disease detection purposes. We pre-select biomarkers to

reduce the number of biomarkers considered to be more practical, reduce overfitting, and to understand the causal effect of the set of biomarkers. With respect to *Sen*@90, and *Sen*@80, the XGB model with causality measure performed better when compared to the baseline for 3 biomarkers and has a hike of 0.114 on AUC. We found that approaches with the causal metric performed better when handling a smaller number of biomarkers, while conventional techniques like univariate feature selection performed better with a larger number of biomarkers. The causality measure compares co-occurring biomarkers, they could provide biological intuition enabling further empirical studies. We see evidence that this approach likely generalizes for the prediction of other diseases based on biomarkers, as our machine learning methods perform well across a variety of diseases.

## Supporting information

**S1 Table. Hyperparameters used for each model**
(PDF)

**S2 Table. Test AUC of various methods for feature selection.**
(PDF)

**S3 Table. Univariate selection with min-max scaling.**
(PDF)

**S4 Table. Most frequent top 3 biomarkers for univariate selection and causal measure.**
(PDF)

**S5 Table. Most frequent top 10 biomarkers for univaritate selection and causal measure.**
(PDF)

**S1 Fig. Confusion matrix for XGB model for 3 biomarkers at threshold 1.4 with the causal metric.**
(TIF)

**S2 Fig. Confusion matrix for GBT model for 10 biomarkers with the univariate method.**
(TIF)

**S3 Fig. ROC Curve with 10% confidence interval for XGB model with causality measure (3 Biomarkers).**
(TIF)

**S4 Fig. ROC Curve with 10% confidence interval for LR model with univariate selection (3 Biomarkers).**
(TIF)

**S5 Fig. ROC Curve with 10% confidence interval for GBT model with univariate selection (10 Biomarkers).**
(TIF)

**S6 Fig. ROC Curve with 10% confidence interval for LR model with univariate selection (10 Biomarkers).**
(TIF)

**S7 Fig. Increase in probability of cases (cancer) with the presence of causal-based biomarkers as a function of *K* with 10% confidence interval.**
(TIF)

## Author contributions

**Conceptualization:** Divyagna Bavikadi, Paulo Shakarian.

**Data curation:** Yunro Chung, Lusheng Song, Ji Qiu.

**Formal analysis:** Divyagna Bavikadi, Paulo Shakarian.

**Investigation:** Divyagna Bavikadi, Yunro Chung, Lusheng Song, Ji Qiu, Paulo Shakarian.

**Methodology:** Divyagna Bavikadi, Ayushi Agarwal, Shashank Ganta, Paulo Shakarian.

**Project administration:** Divyagna Bavikadi.

**Software:** Divyagna Bavikadi, Shashank Ganta.

**Validation:** Divyagna Bavikadi, Ayushi Agarwal, Shashank Ganta.

**Visualization:** Divyagna Bavikadi, Ayushi Agarwal, Shashank Ganta.

**Writing – original draft:** Divyagna Bavikadi, Paulo Shakarian.

**Writing – review & editing:** Divyagna Bavikadi.

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
