## [Decision Letter · Decision Letter 0]

12 Jul 2024

PONE-D-24-21087Machine Learning Driven Biomarker Selection for Medical DiagnosisPLOS ONE

Dear Dr. Shakarian,

Thank you for submitting your manuscript to PLOS ONE. After careful consideration, we feel that it has merit but does not fully meet PLOS ONE’s publication criteria as it currently stands. Therefore, we invite you to submit a revised version of the manuscript that addresses the points raised during the review process.

We look forward to receiving your revised manuscript.

Kind regards,

John Adeoye

Academic Editor

PLOS ONE

Journal Requirements:

4. Thank you for uploading your study's underlying data set. Unfortunately, the repository you have noted in your Data Availability statement does not qualify as an acceptable data repository according to PLOS's standards.

Reviewers' comments:

Reviewer's Responses to Questions

**Comments to the Author**

1. Is the manuscript technically sound, and do the data support the conclusions?

Reviewer #1: Partly

Reviewer #2: Yes

2. Has the statistical analysis been performed appropriately and rigorously? 

Reviewer #1: I Don't Know

Reviewer #2: Yes

3. Have the authors made all data underlying the findings in their manuscript fully available?

Reviewer #1: Yes

Reviewer #2: Yes

4. Is the manuscript presented in an intelligible fashion and written in standard English?

Reviewer #1: Yes

Reviewer #2: Yes

5. Review Comments to the Author

**Reviewer #1:** 1. Confusion Matrix is missing

2. Discussion part is weak .

3. Comparison with state-of-the-art studies is missing.

4. Detailed diagram of the whole work is missing.

5. Perform some statistical testing to justify the superiority of the work .

6. The cited literature is not updated, it is recommended to add relevant literature from the past three years. By using relatively new literature as support, it is helpful to reflect the progressiveness of this model. Moreover, this paper applies feature selection methods in the medical field and suggests adding feature selection methods that are more relevant to medical image recognition. And in the process of discussion, it is suggested that the author classify and summarize these literature reasonably. The related work is not simply listing recent situations. Suggest the author to analyze these feature selection methods in detail. Analyze the advantages and disadvantages of these methods, and add textual descriptions to connect these listed literature. In addition, after analyzing these related works, it is recommended that the author summarize the current situation in this field and propose the unresolved issues in this field at the current stage. It is worth noting that these issues are addressed by the model proposed in this paper.

7. These few recent state-of-the-art studies should be discussed where feature selection for human disease recognition is solved efficiently (Nature-Inspired Algorithms-Based Optimal Features Selection Strategy for COVID-19 Detection Using Medical Images;An enhanced efficient approach for feature selection for chronic human disease prediction: A breast cancer study;An enhanced soft-computing based strategy for efficient feature selection for timely breast cancer prediction: Wisconsin Diagnostic Breast Cancer dataset case;A novel approach for human diseases prediction using nature inspired computing & machine learning approach;)

Best Regards

**Reviewer #2: **This reviewer would like to thank the authors for a well-written article seeking to illustrate the usefulness of feature selection methods and machine learning methods when looking to identify biomarkers for potential medical diagnosis.

There are a few areas that need additional information. The dataset used is one from gastric cancer. However, many of the studies referenced are from very different areas (and not cancer at all). It might be nice to also include previous studies/publications using machine learning and comparing it to logistic regression on gastric cancer/h. pylori studies to make the work done here a little more relevant to prior work in the field.

It also might be nice to reference prior analyses done (by yourself or others) with the gastric cancer dataset under investigation, so that readers can get a sense of whether your methods confirm/differ with what is currently known in that specific field.

There is no limitations section included in the manuscript. This reviewer thinks one would be advisable. Such things to include might be the relatively small sample size (only 100 subjects in the dataset used, but a rather large number of potential parameters) and the impact that might have on ML methods. The implications for overfitting the data since that is mentioned in one of the sections, the lack of clinical data in the dataset and how that might make medical diagnosis hard from an initial clinic visit, justification for thresholds for the protein data, etc.

The authors mention potentially binarizing the biomarker inputs. They should discuss exactly how the different threshold values were arrived at as well as the justification for doing so. In the field of biostatistics, it is generally not advisable to categorize data (make it binary) without sound justification. This is lacking here and should be added so readers understand how and why this was done (and the implications for doing so added to a limitations section). The text mentions that the data values range from 0 to 260, so having a threshold maximum of 1,8 seems a bit odd.

How were the top 3 and top 10 biomarkers actually selected? lowest p-value for the chi-square test? What metric for the causal method (lowest, highest value? something else? If p-value, how was that obtained?)?

What software packages were used for logistic regression, random forest, other methods not mentioned in the text? This information should be included for reproducibility purposes. Also, were function defaults used (for RF, example how many trees were allowed to be built, how many variables were allowed to be examined at each split, etc) or were other options selected? This should be mentioned as well. Additionally, how were the training and testing sets calculated (and what was the relative size of each 80/20 split, something else)?

For the Tables, please include mention of the various abbreviations used in the table legend at the bottom. The underline is hard to see - consider some other way to distinguish this.

For the discussion section, it might be nice to compare/contrast the results found here (especially pertaining to the 3 markers mentioned as potentially interesting) and how those compare to prior studies using this dataset. Are they in agreement? Disagreement? Similarly, how do these three markers compare with other work in the realm of gastric cancer? Are they new? Are they in agreement with previous work?

6. PLOS authors have the option to publish the peer review history of their article (what does this mean?). If published, this will include your full peer review and any attached files.

Reviewer #1: No

Reviewer #2: No

---

## [Author Response · Author response to Decision Letter 1]

10 Nov 2024

Thank you very much for the suggestions and insightful comments into the field. We address each comment as follows:

Reviewer 1

1. We added the confusion matrix in the appendix for added performance results as figure s1. Kindly note that, AUC, ROC are more relevant in this application as they show the precision and recall tradeoff for all thresholds for the predicted probabilities and so is Sen@90 to evaluate the model that ensures in identifying most disease cases. Hence, we included those in the main paper. On the other hand, the confusion matrix reports values when a particular threshold for the predicted probabilities (=0.5) is used for classification with limited insight on the overall model performance.

2. We added a secondary analysis in the second paragraph in the discussion section assessing the increase in probability of cases based on the presence of most frequent biomarkers and also talk about future evaluation. We also talk about limitations and effect of the binarization experiments.

3. We previously included a widely applied feature selection method- univariate selection. Please note that as added in line 61-66, we chose this baseline feature selection method as it’s more applicable in this case.

4. Thank you for your comment. We added the overall workflow diagram in the second paragraph of Overview of approaches subsection in Machine Learning and Feature Selection Methods section.

5. Standard with cross validation [1,2] based ML approaches, to validate our approach, we revise two assumptions that violate most statistical tests-

1. We have only 1 fold, that is not large enough population to apply standard statistical tests. 2. Each fold is not independent from each other as they are produced from the same dataset.

That said, in order to communicate the significance of our results we also provide the error bars based on confidence bands on 10% error for figure 2,3,4,5.

6. We have modified the second paragraph in the related work accordingly and noted the differences in the literature to our approach. Note that medical images inherently have different attributes than protein biomarker measures and medical image recognition is not the focus of our work. We have added some more literature from line 51-66 in the second paragraph of the related works section.

7. Second paragraph in related work section is added, we also clarified that our results are not about these diagnosis in general but narrow area of disease diagnosis with the use of biomarkers with a causality measure. We clarify this and talk about how this work is positioned vis a vis the papers that the reviewer mentioned in line 53,77 in the second paragraph of the related work section and we thank the reviewer again for this insightful comment.

Reviewer 2

1. In addition to already mentioned work, we added more cancer related work as pointed out in lines 53,58,60,75.

2. We added the line 107 in Gastric Cancer Dataset section to point out that this kind of work is not explored in this particular dataset. Also, expanded our discussion in the related work to show similar work and analysis done on gastric cancer datasets.

3. The main limitation as mentioned before is for higher number of biomarkers, we add a third paragraph in the discussion section to discuss limitations about implications for overfitting, and added the fourth paragraph in the discussion section to discuss the binarized data experiments.

4. The reason for setting those threshold values is added in the last line of the first paragraph of Overview of approaches subsection in Machine Learning and Feature Selection Methods section. The justification of binarized data experiments is mentioned in the fourth paragraph in the discussion section to discuss the binarized data experiments.

5. Thanks for pointing it out, we clarify that by adding a sentence in 4th line of the second paragraph of Overview of approaches subsection in Machine Learning and Feature Selection Methods section.

6. The hyperparameter details previously mentioned in appendix is moved and pointed out in the main paper in the second paragraph of the setup subsection of the results section. We also added the software packages used.

7. Thank you for your comment, we updated the legend to include asterisk and mentioned the abbreviation beforehand in the paragraph of Machine learning classification methods subsection in Machine Learning and Feature Selection Methods section.

8. We added secondary analysis to show the causal relationship of biomarkers in the second paragraph of the discussion section for this dataset. In order to compare the biomarkers with other work in the gastric cancer realm, it would have to be in a dataset where atleast all analytes with this dataset are included. We also noted that in order to fairly compare the biomarkers found, we need to conduct further clinical tests pertaining to those specific biomarkers.

References:

1. Cawley, Gavin C.; Talbot, Nicola L. C. (2010). "On Over-fitting in Model Selection and Subsequent Selection Bias in Performance Evaluation" . Journal of Machine Learning Research. 11: 2079–2107.

2. Grossman, Robert; Seni, Giovanni; Elder, John; Agarwal, Nitin; Liu, Huan (2010). "Ensemble Methods in Data Mining: Improving Accuracy Through Combining Predictions". Synthesis Lectures on Data Mining and Knowledge Discovery. 2. Morgan & Claypool: 1–126. doi:10.2200/S00240ED1V01Y200912DMK002.

---

## [Decision Letter · Decision Letter 1]

28 Nov 2024

PONE-D-24-21087R1Machine learning driven biomarker selection for medical diagnosisPLOS ONE

Dear Dr. Bavikadi,

Thank you for submitting your manuscript to PLOS ONE. After careful consideration, we feel that it has merit but does not fully meet PLOS ONE’s publication criteria as it currently stands. Therefore, we invite you to submit a revised version of the manuscript that addresses the points raised during the review process.

We look forward to receiving your revised manuscript.

Kind regards,

John Adeoye

Academic Editor

PLOS ONE

Reviewers' comments:

Reviewer's Responses to Questions

**Comments to the Author**

1. If the authors have adequately addressed your comments raised in a previous round of review and you feel that this manuscript is now acceptable for publication, you may indicate that here to bypass the “Comments to the Author” section, enter your conflict of interest statement in the “Confidential to Editor” section, and submit your "Accept" recommendation.

Reviewer #2: (No Response)

2. Is the manuscript technically sound, and do the data support the conclusions?

Reviewer #2: Yes

3. Has the statistical analysis been performed appropriately and rigorously? 

Reviewer #2: No

4. Have the authors made all data underlying the findings in their manuscript fully available?

Reviewer #2: Yes

5. Is the manuscript presented in an intelligible fashion and written in standard English?

Reviewer #2: Yes

6. Review Comments to the Author

Reviewer #2: This reviewer thanks the authors for their initial responses to prior reviewer comments. I have additional comments that can better help the reader understand the information presented.

The "Related Work" section is somewhat confusing, as presented. By the time the reader gets to that section, they do not yet know the exact methods that the authors will be employing to analyze the data. Thus, it is confusing to read about related work talking about various methods, when we don't know methods yet. I suggest either introducing the exact methods used before this section (and talking a bit about your process), OR moving this section to after the methods description.

The description of the causal measure (in the methods section) is a bit dense and not easy for the non-math/technical person to fully grasp. Maybe providing an example dataset (small) and how the causal (i) is calculated would help.

In the ML classification second, is it 4 or 5 methods that are used? The text says 4, but then describes 5. What does DNN stand for? Also, RELU? You should expand all acronyms, at least the first use of them.

It is unclear why chi-square tests were used for the univariate analysis. The data, as it appears in the supplemental material section, appears to be continuous. Mann-Whitney or t-tests would be more appropriate for continuous data. Chi-square tests are used for count data and it is not clear that is what is obtained. Also, since there were so many tests run, were multiple hypothesis corrections run? If not, why not?

The confidence intervals presented are 10% and not the more common 5%. Why was 10% chosen?

Please state the size of the training and testing sets used.

It would also be nice to see the results of the univariate & causal tests in the supplemental materials. Would be also nice to see the lists of the 3 & 10 BM panels used in each scenario (again in supplemental).

Did you consider scaling your data for any of your metrics?

The data was binarized for various situations. Please expand on the limitations of doing this, especially pertaining to choice of a split-point for doing so.

The full dataset was included in the supplemental materials. Please consider reposing it somewhere on the web so that other individuals could actually use it easily.

7. PLOS authors have the option to publish the peer review history of their article (what does this mean?). If published, this will include your full peer review and any attached files.

Reviewer #2: No

---

## [Author Response · Author response to Decision Letter 2]

30 Jan 2025

Thank you for your insightful comments and suggestions for clarifications. We address each comment as follows:

Reviewer #2:

1. We moved the ‘Related Work’ section to a last second section, before the final section ‘Conclusion’ for clarity, so that the reader will be aware of the employed methods when reading the related work.

2. We added an example of causal metric computation on a sample dataset along with a table (table 2) in the machine learning and feature selection methods section. We also modified the description of the derivation of the causal metric in S2 Appendix to make it more clear.

3. Yes, it is 5. We fixed the typo in the Introduction section and Abstract. We updated the text to elaborate all abbreviations in the first use in page 5.

4. Thank you for your comment. We primarily used chi-square statistic based univariate feature selection as the labels in the dataset and provided by the models is discrete (binary) and in many cases we discretize the data in intermediate steps, for instance in computation of the causal measure. Also, we report results for discretized data for the univariate selection as well. However, we also experimented with statistic typically used for continuous values like both Mann-Whitney and t-tests, where it resulted in comparable or lower test AUCs. Mann-Whitney test gave lower test AUC values than all other baselines. The t-test gave test AUCs comparable to the chi-square test with a difference in performance on average is 0.01. However, for 3 biomarkers, MLP with causal method still outperforms all baselines and for 10 biomarkers, MLP with chi-square is still the best performing model as reported in the main experimental results. We include these new findings in the appendix S3, Additional tests para.

We did try multiple hypothesis correction, more specifically, the Benjamini-Hochberg (BH) procedure. It provided superior results in limited number of settings. Although, we didn’t include it in the main paper as on average, it did not result in consistent significant improvements in the AUCs but we added them in the appendix S3, Additional tests para.

5. Thank you for pointing it out. We have results for 5% as well and we added them to the main paper as it is more standard. We added the results with 10% confidence interval in the appendix S3, Main Results with 10% confidence intervals para.

6. As mentioned in the setup subsection of the results section as well as in second paragraph of overview of approaches, we employed standard leave-one-out cross-validation (LOOCV). For a dataset of the size N, we do N iterations of train and test splits with the test size of 1 as standard with LOOCV in machine learning methods.

7. Thank you for the suggestion. We updated the main manuscript with a further elaborate figure of top 10 most frequent biomarkers over all folds for various settings of thresholds for a better view of selected biomarkers across all settings. Additionally, we also added the resulting biomarkers picked for the best performing models as a table in the appendix S3, Results of selected top K biomarkers para.

8. Yes, we did try other scaling methods like min-max scaling for the baseline of univariate selection. We add this result to the appendix S3, Experiment with min-max scaling para. This resulted in an average decline of 0.14 in the test AUCs when compared to the baseline reported in the main results section. Also, as the causal method inherently uses thresholding, scaling the data here wouldn’t significantly impact the results due to the intermediate step of thresholding (when the measurement is above the threshold, the data is considered to be 1 or 0 otherwise).

9. We updated the first para in hyperparameter study to discuss more on the effect of thresholds for binarizing the data. Generally, the threshold 1 seems to give lower test AUCs and the threshold 1.4 is the most optimal. Also, as noted in the final para’s 5th line (line 243), binarized experiments perform in a similar fashion as the non-binarized ones.

---

## [Decision Letter · Decision Letter 2]

25 Mar 2025

Machine learning driven biomarker selection for medical diagnosis

PONE-D-24-21087R2

Dear Dr. Bavikadi,

We’re pleased to inform you that your manuscript has been judged scientifically suitable for publication and will be formally accepted for publication once it meets all outstanding technical requirements.

Kind regards,

John Adeoye

Academic Editor

PLOS ONE

Additional Editor Comments (optional):

Reviewers' comments:

Reviewer's Responses to Questions

**Comments to the Author**

1. If the authors have adequately addressed your comments raised in a previous round of review and you feel that this manuscript is now acceptable for publication, you may indicate that here to bypass the “Comments to the Author” section, enter your conflict of interest statement in the “Confidential to Editor” section, and submit your "Accept" recommendation.

Reviewer #2: All comments have been addressed

2. Is the manuscript technically sound, and do the data support the conclusions?

Reviewer #2: (No Response)

3. Has the statistical analysis been performed appropriately and rigorously? 

Reviewer #2: (No Response)

4. Have the authors made all data underlying the findings in their manuscript fully available?

Reviewer #2: (No Response)

5. Is the manuscript presented in an intelligible fashion and written in standard English?

Reviewer #2: (No Response)

6. Review Comments to the Author

Reviewer #2: (No Response)

7. PLOS authors have the option to publish the peer review history of their article (what does this mean?). If published, this will include your full peer review and any attached files.

Reviewer #2: No

---

## [Editor Report · Acceptance letter]

PONE-D-24-21087R2

PLOS ONE

Dear Dr. Bavikadi,

I'm pleased to inform you that your manuscript has been deemed suitable for publication in PLOS ONE. Congratulations! Your manuscript is now being handed over to our production team.

Kind regards,

on behalf of

Dr. John Adeoye

Academic Editor

PLOS ONE